# Learning Interface Conditions in Domain Decomposition Solvers

**Ali Taghibakhshi**
Mechanical Science and Engineering
University of Illinois Urbana-Champaign
Urbana, IL 61801, USA
alit2@illinois.edu

**Nicolas Nytko**
Computer Science
University of Illinois Urbana-Champaign
Urbana, IL 61801, USA
nnytko2@illinois.edu

**Tareq Zaman**
Scientific Computing Program
Memorial University of Newfoundland
and Labrador
St. John's, NL, Canada
tzaman@mun.ca

**Scott MacLachlan**
Mathematics and Statistics
Memorial University of Newfoundland
and Labrador
St. John's, NL, Canada
smaclachlan@mun.ca

**Luke Olson**
Computer Science
University of Illinois Urbana-Champaign
Urbana, IL 61801, USA
lukeo@illinois.edu

**Matthew West**
Mechanical Science and Engineering
University of Illinois Urbana-Champaign
Urbana, IL 61801, USA
mwest@illinois.edu

## Abstract

Domain decomposition methods are widely used and effective in the approximation of solutions to partial differential equations. Yet the *optimal* construction of these methods requires tedious analysis and is often available only in simplified, structured-grid settings, limiting their use for more complex problems. In this work, we generalize optimized Schwarz domain decomposition methods to unstructured-grid problems, using Graph Convolutional Neural Networks (GCNNs) and unsupervised learning to learn optimal modifications at subdomain interfaces. A key ingredient in our approach is an improved loss function, enabling effective training on relatively small problems, but robust performance on arbitrarily large problems, with computational cost linear in problem size. The performance of the learned linear solvers is compared with both classical and optimized domain decomposition algorithms, for both structured- and unstructured-grid problems.

## 1 Introduction

Domain decomposition methods (DDMs) [1, 2, 3] are highly effective in solving the linear and nonlinear systems of equations that arise from the numerical approximation of solutions to partial differential equations (PDEs). While most effective on elliptic boundary-value problems, DDMs can also be applied to nonlinear problems, either using their nonlinear variants, or successively solving linearizations. Time-dependent problems are normally solved by using a time stepping algorithm in the time domain for implicit methods, which require the solution of a spatial problem for each time step. Of these methods, Schwarz methods are particularly popular given their relative simplicity and ease of parallelization. The common theme is to break the global problem into subproblems, derived either by projection or by discretizing the same PDE over a physical subdomain, and to use solutions

36th Conference on Neural Information Processing Systems (NeurIPS 2022).

on the subdomains as a preconditioner for the global discretization. Classical Schwarz methods generally make use of Dirichlet or Neumann boundary conditions for these subdomain problems, while Optimized Schwarz Methods (OSMs) aim to improve the convergence of the algorithm by using more general interface conditions [4]. Notably, [5] demonstrates that optimal, but non-local, interface conditions exist for more general decompositions.

Much of the OSM literature considers only one-level additive Schwarz methods, although multilevel extensions do exist. For one-level methods (i.e., domain decomposition approaches without a "coarse grid"), restricted additive Schwarz (RAS) approaches [6] are arguably the most common; optimized restricted additive Schwarz (ORAS) methods are considered in [7]. The OSM idea has also been extended to asynchronous Schwarz methods [8], where the computations on each subdomain are done using the newest information available in a parallel computing environment without synchronizing the solves on each subdomain.

With a recent focus on machine learning (ML) techniques for solving PDE systems [9, 10], there is also effort to apply learning-based methods to improve the performance of iterative solvers for PDEs, including DDM and algebraic multigrid (AMG) methods. Within AMG methods, ML techniques have been applied to learning interpolation operators [11, 12] and to coarse-grid selection in reduction-based AMG [13]. Of particular note here is the loss function employed in [11, 12], where they use unsupervised learning to train a graph neural network to minimize the Frobenius norm of the error-propagation operator of their iterative method. Within DDM, significant effort has been invested in combining ML techniques with DDM, as in [14], where two main families of approaches are given: 1) using ML within classical DDM methods to improve convergence, and 2) using deep neural networks, such as Physics Informed Neural Networks (PINNs) [9], as a discretization module and solver for DDM problems. In [15], a fully connected neural network is used to predict the geometric locations of constraints for coarse space enhancements in an adaptive Finite Element Tearing and Interconnecting-Dual Primal (FETI-DP) method. Using the continuous formulation of DDM, the so-called D3M [16] uses a variational deep learning solver, implementing local neural networks on physical subdomains in a parallel fashion. Likewise, Deep-DDM [17] utilizes PINNs to discretize and solve DDM problems, with coarse space corrections [18] being used to improve scalability.

In this paper, we advance DDM-based solvers by developing a framework for learning optimized Schwarz preconditioners. A key aspect of this is reconsidering the loss function to use a more effective relaxation of the ideal objective function than Frobenius norm minimization [11, 12]. Moreover, the approach introduced here offers an opportunity to reconsider existing limitations of optimized Schwarz methods, where optimal parameter choice is based on Fourier analysis and requires a highly regular subdomain structure, such as in the classical cases of square domains split into two equal subdomains or into regular grids of square subdomains. Our framework learns the optimized Schwarz parameters via training on small problem sizes, in a way that generalizes effectively to large problems, and in a way that allows us to consider both irregular domains and unstructured grids, with no extraordinary limitations on subdomain shape. Furthermore, the evaluation time of our algorithm scales linearly with problem size. This allows significant freedom in defining optimized Schwarz methods, in comparison to classical approaches, allowing us to explore the potential benefits of optimized Schwarz over classical (restricted) additive methods on unstructured grids for the first time.

## 2 Background

Let $\Omega \subset \mathbb{R}^2$ be an open set, and consider the positive-definite Helmholtz problem

$$Lu = (\eta - \Delta)u = f \quad \text{in } \Omega, \tag{1}$$

with inhomogeneous Dirichlet conditions imposed on the boundary $\partial\Omega$. In (1), the parameter $\eta > 0$ represents a *shift* in the Helmholtz problem. In the numerical results below, we consider both finite-difference discretizations of (1) on regular grids, as well as piecewise linear finite-element (FE) discretizations on arbitrary triangulations. In both cases, we denote the set of degrees of freedom as $D$, and note that these are in a one-to-one correspondence with the nodes of the underlying mesh. Consider a decomposition of $D$ into non-overlapping subdomains $D_i^0, i \in \{1, 2, \ldots, S\}$ such that each node is contained within exactly one subdomain $D_i^0$, yielding $\cup D_i^0 = D$. In this subdomain notation, the superscript denotes the amount of overlap in the subdomains, which is zero for the non-overlapping subdomains that we first consider. Let $R_i^0$ be the restriction operator onto the set of degrees of freedom (DoFs) in $D_i^0$, and let $\left(R_i^0\right)^T$ be the corresponding extension operator from $D_i^0$

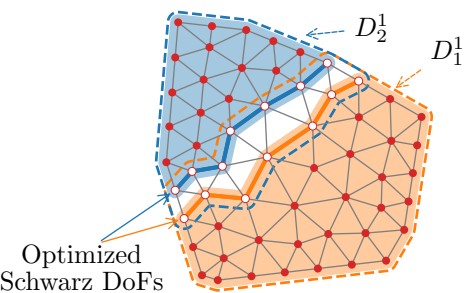

Figure 1: Two subdomains with overlap $\delta = 1$ on a 58-node unstructured grid. The blue and orange shading denotes the original non-overlapping partitions ($D_1^0$ and $D_2^0$), while the blue and orange dashed outlines show the overlapping subdomains ($D_1^1$ and $D_2^1$). Nodes belonging to only one subdomain are marked with a solid circle, while those in white belong to both subdomains. The connections in the boundary matrices $L_1$ and $L_2$ are denoted by the edges shaded in blue (for $L_1$) and orange (for $L_2$).

into set $D$. Then, an FE discretization of the Helmholtz problem leads to a linear system of the form $Ax = b$, where $A$ is the global stiffness matrix and $A_i^0 = R_i^0 A \left(R_i^0\right)^T$ is the subdomain stiffness matrix for $D_i^0$. We note that alternate definitions to the Galerkin projection for $A_i^0$ are possible, and are commonly considered in optimized Schwarz settings (as noted below).

In the case of restricted additive Schwarz (RAS) [6], the subdomains are extended to allow for overlap: nodes near the "boundary" of their subdomain are potentially included in two or more subdomains. We denote the amount of overlap by $\delta \in \mathbb{N}$, defining subdomains $D_i^\delta$ recursively, by $D_i^\delta = D_i^{\delta-1} \cup \left\{j \mid a_{kj} \neq 0 \text{ and } k \in D_i^{\delta-1}\right\}$ for $\delta > 0$. The conventional RAS preconditioner is defined as

$$M_{\text{RAS}} = \sum_{i=1}^S (\tilde{R}_i^\delta)^T (A_i^\delta)^{-1} R_i^\delta, \tag{2}$$

where $R_i^\delta$ is the standard restriction operator to subdomain $\Omega_i^\delta$, $\tilde{R}_i^\delta$ is a modified restriction operator from $D$ to the DoFs in $D_i^\delta$ that takes nonzero values only for DoFs in $D_i^0$, and $A_i^\delta = (R_i^\delta)^T A R_i^\delta$. Figure 1 shows an example unstructured grid with two subdomains and overlap $\delta = 1$.

In an optimized Schwarz setting, we modify the subdomain systems, $A_i^\delta$. Rather than using a Galerkin projection onto $D_i^\delta$, we rediscretize (1) over the subdomain of $\Omega$ corresponding to these DoFs, imposing a Robin-type boundary condition on the boundary of the subdomain. We define this matrix to be $\tilde{A}_i^\delta = A_i + L_i$, where $A_i$ is the term resulting from discretization of (1) with Neumann boundary conditions, and $L_i$ is additional the term resulting from the Robin-type condition, as in:

$$\text{Dirichlet: } u = g_{\text{D}}(x), \quad \text{Neumann: } \vec{n} \cdot \nabla u = g_{\text{N}}(x), \quad \text{Robin: } \alpha u + \vec{n} \cdot \nabla u = g_{\text{R}}(x), \tag{3}$$

where $\vec{n}$ is the outward normal to the edge on the boundary and $g$ denotes inhomogeneous data.

The matrix $L_i$ has the same dimensions as $A_i$, so that $\tilde{A}_i^\delta$ is well-defined. However, it has a significantly different sparsity pattern, with nonzero entries only in rows/columns corresponding to nodes on the boundary of subdomain $D_i^\delta$. In practice, we identify a cycle or path in the graph corresponding to $A_i$ with the property that every node in the cycle is on the boundary of $D_i^\delta$ but not the boundary of $D$ (the discretized domain), and then restrict the nonzeros in $L_i$ to the entries corresponding to the edges in this cycle/path (including self-edges, corresponding to entries on the diagonal of $L_i$).

Using this notation, the ORAS preconditioner can be written as

$$M_{\text{ORAS}} = \sum_{i=1}^S \left(\tilde{R}_i^\delta\right)^T \left(\tilde{A}_i^\delta\right)^{-1} R_i^\delta. \tag{4}$$

Our aim is to learn the values in the matrices $L_i$, aiming to outperform the classical choices of these values predicted by Fourier analysis, in the cases where those values are known, and to learn

suitable values for cases, such as finite-element discretizations on unstructured grids, where no known optimized Schwarz parameters exist. We optimize the values for the case of stationary (Richardson) iteration, but evaluate the performance of the resulting methods both as stationary iterations and as preconditioners for FGMRES.

**Graph Neural Networks (GNNs):** Applying learning techniques to graph structured data necessitates stepping beyond multilayer perceptron (MLP) and conventional convolutional neural networks (CNN) to a type of network that leverages the underlying graph nature of the problem, namely graph convolutional neural networks (GCNNs). GCNNs are typically divided into two categories: spectral and spatial [19]. Spectral GCNNs, first introduced by Bruna et al. [20], consider a graph Laplacian eigenbasis and define convolution as diagonal operators. As such, spectral GCNN methods suffer from time complexity problems due to the necessity for the eigendecomposition of the Laplacian matrix. Nevertheless, in follow-up work [21, 22], remedies have been proposed to mitigate this. Unlike spectral methods, spatial GCNNs consider local propagation of information in graph domains as a convolution graph. One popular framework is the message passing neural network (MPNN) [23], which is based on sharing information among neighbor nodes in each round of a convolution pass. This has been generalized [24] by introducing a configurable graph network block structure consisting of node and edge convolution modules and a global attribute. In an effort to alleviate computational complexity of GCNNs [25], topology adaptive graph convolution networks (TAGCN) can be constructed by defining learnable filters. This is not only computationally simpler, but also allows for adapting to the topology of the graphs when scanning them for convolution.

## 3 Method

### 3.1 Optimization problem and the loss function

Throughout this paper, we use $\|\cdot\|$ to denote the $\ell^2$ norm of a matrix or vector, $\|A\|_F$ for the Frobenius norm of $A$, and $\rho(A)$ as the spectral radius of $A$. The optimization problem that we seek to solve is to find optimal values for the entries in the matrices $L_i$, constrained by given sparsity patterns, to minimize $\rho(T)$, where $T = I - M_{\text{ORAS}}A$ is the error-propagation operator for the stationary iteration corresponding to $M_{\text{ORAS}}$ defined in (4). The spectral radius $\rho(T)$ corresponds to the *asymptotic convergence factor* of the stationary iteration, giving a bound on the asymptotic convergence of the method. Formally defined in terms of the extremal eigenvalue of $T^T T$ (since $T$ is not symmetric), direct minimization of $\rho(T)$ is difficult since backpropagation of an eigendecomposition is numerically unstable [26]. To overcome this, Luz et al. [12] propose to relax the minimization of $\rho(T)$ (for a similar AMG preconditioner) to minimizing the Frobenius norm of $T$, $\|T\|_F$. In our experiments, however, we find that this is insufficient, leading to preconditioners that do not scale. One reason is that while the Frobenius norm is an upper bound on the spectral radius, it does not appear to be a suitably "tight" bound for use in this context (see Section 4.2 and Figure 6). Instead, we use a relaxation inspired by Gelfand's formula, that $\rho(T) = \lim_{K \to \infty} \|T^K\|^{\frac{1}{K}}$, and the common bound that

$$\rho(T) \le \|T^K\|^{\frac{1}{K}} = \sup_{x \neq 0} \left( \frac{\|T^K x\|}{\|x\|} \right)^{\frac{1}{K}} = \sup_{x:\|x\|=1} (\|T^K x\|)^{\frac{1}{K}} \tag{5}$$

for some finite $K \in \mathbb{N}$. This results in the optimization problem

$$\min_{\substack{L_i, i=1,2,\ldots,S \\ \text{sparsity of } L_i}} \sup_{x:\|x\|=1} \|T^K x\|. \tag{6}$$

### 3.2 Numerical evaluation of the loss function

We denote the action of evaluating the GNN by $f^{(\theta)}$ (where $\theta$ represents the network parameters), and consider a discretized problem with DoF set $D$, of size $n$. The set $D$ can be decomposed into subdomains either by using fixed geometric choices of the subdomain (e.g., for finite-difference discretizations), using the METIS graph partitioner [27], or a $k$-means-based clustering algorithm (best known as Lloyd's algorithm which has $O(n)$ time complexity) [28, 29]. For unstructured problems, we use a $k$-means-based algorithm, decomposing $D$ to subdomains $D_i$ for $i = 1, 2, \ldots, S$, with overlap $\delta$; see Supplementary Materials for details. The GNN then takes $D$ and its decomposition

as inputs, as well as sparsity constraints on the matrices $L_i$ for $i = 1, 2, \ldots, S$, and outputs values for these matrices:

$$L_1^{(\theta)}, L_2^{(\theta)}, \ldots, L_S^{(\theta)} \leftarrow f^{(\theta)}(D). \tag{7}$$

Using the learned subdomain interface matrices, we then obtain the modified MLORAS (Machine Learning Optimized Restricted Additive Schwarz) operator, $M_{\mathrm{ORAS}}^{(\theta)}$, simply using $\tilde{A}_i^\delta = A_i + L_i^{(\theta)}$ in (4). We denote the associated error propagation operator by $T^{(\theta)} = I - M_{\mathrm{ORAS}}^{(\theta)} A$.

While Gelfand's formula and the associated upper bound in (5) are valid in any norm, it is natural to consider them with respect to the $\ell^2$ norm in this setting. However, this raises the same issue as encountered in [12], that it generally requires an eigendecomposition to compute the norm. To avoid this, we use a stochastic sampling of $\left\| \left( T^{(\theta)} \right)^K \right\|$, generated by the sample set $X \in \mathbb{R}^{n \times m}$ for some $m \in \mathbb{N}$, given as

$$X = [x_1, x_2, \ldots, x_m], \forall_j \ x_j \sim \mathbb{R}^n \text{ uniformly}, \|x_j\| = 1. \tag{8}$$

Here, we randomly select $m$ points uniformly on a unit sphere in $\mathbb{R}^n$, which can be done using the method in [30]. We then define

$$Y^{(\theta)} = \left\{ \left\| \left( T^{(\theta)} \right)^K x_1 \right\|, \left\| \left( T^{(\theta)} \right)^K x_2 \right\|, \ldots, \left\| \left( T^{(\theta)} \right)^K x_m \right\| \right\}, \tag{9}$$

taking each column of $X$ as the initial guess to the solution of the homogeneous problem $Ax = 0$ and taking $K$ steps of the stationary algorithm to generate $\left( T^{(\theta)} \right)^K x_j$. Since we normalize each column of $X$ to have $\|x_j\| = 1$, the value of $\left\| \left( T^{(\theta)} \right)^K x_j \right\|$ serves as a lower bound for $\left\| \left( T^{(\theta)} \right)^K \right\|$. Thus, taking the maximum of the values in $Y$ provides a practical loss function that we use below, defining

$$\mathcal{L}^{(\theta)} = \max(Y^{(\theta)}). \tag{10}$$

We note similarities between this loss function and that used in [31], but that we are able to use the maximum of $Y^{(\theta)}$ (giving a better approximation to the norm) in our context instead of averaging.

Ultimately, the cost of our algorithm depends strongly on the chosen values of $m$ and $K$. For sufficiently large values of $m$, we now show that the maximum value in $Y^{(\theta)}$ is an arbitrarily good approximation to $\left\| \left( T^{(\theta)} \right)^K \right\|^{\frac{1}{K}}$ (in the statistical sense).

**Theorem 1.** *For any nonzero matrix $T$, $\epsilon > 0$, and $\delta < 1$, there exist $M, K \in \mathbb{N}$ such that for any $m > M$, if one chooses $m$ points, $x_j$, uniformly at random from $\{x \in \mathbb{R}^n, \|x\| = 1\}$, then $Y = \left\{ \left\| T^K x_1 \right\|, \left\| T^K x_2 \right\|, \ldots, \left\| T^K x_m \right\| \right\}$ satisfies*

$$P \left( \left| \rho(T) - \max(Y)^{\frac{1}{K}} \right| < \epsilon \right) > 1 - \delta. \tag{11}$$

*Proof.* According to Gelfand's theorem, there exists $L \in \mathbb{N}$ such that $\forall \ell > L$, $\left| \rho(T) - \sup_{x:\|x\|=1} \|T^\ell x\|^{\frac{1}{\ell}} \right| < \frac{\epsilon}{2}$. Take any $K \geq L$ and let $\tilde{\epsilon} = \frac{\epsilon}{2\|T^K\|^{\frac{1}{K}}}$. Since $\mathbb{R}^n$ is finite-dimensional, there exists an $x^* \in \mathbb{R}^n$, $\|x^*\| = 1$ such that $\sup_{x:\|x\|=1} \|T^K x\| = \|T^K x^*\|$. Denote the volume of the surface of the $n$-dimensional sphere of unit radius around the origin in $\mathbb{R}^n$ by $C_{\mathrm{tot}}$, and the volume of the region on this sphere within radius $\tilde{\epsilon}^K$ of $x^*$ by $C_{\tilde{\epsilon}, K}$.

Given $\delta < 1$, let $M \in \mathbb{N}$ satisfy $M \geq \frac{\log(\delta)}{\log\left(1 - \frac{C_{\tilde{\epsilon}, K}}{C_{\mathrm{tot}}}\right)}$. Then,

$$P \left( \|x^* - x_i\| > \tilde{\epsilon}^K \text{ for all } i \right) < \left( 1 - \frac{C_{\tilde{\epsilon}, K}}{C_{\mathrm{tot}}} \right)^M \leq \delta. \tag{12}$$

Thus, with probability of at least $1 - \delta$, we expect at least one point from a selection of $m > M$ points uniformly distributed on the sphere of radius unit radius to be in $C_{\tilde{\epsilon}, K}$. Let that point be $x_r$,

giving

$$\left\| T^K x^* \right\|^{\frac{1}{K}} - \left\| T^K x_r \right\|^{\frac{1}{K}} \leq \left( \left\| T^K x^* \right\| - \left\| T^K x_r \right\| \right)^{\frac{1}{K}} \tag{13}$$

$$\leq \left( \left\| T^K (x^* - x_r) \right\| \right)^{\frac{1}{K}} \tag{14}$$

$$\leq \left\| T^K \right\|^{\frac{1}{K}} \left\| x^* - x_r \right\|^{\frac{1}{K}} \leq \left\| T^K \right\|^{\frac{1}{K}} \tilde{\epsilon} = \frac{\epsilon}{2} \tag{15}$$

using Lemma 3 (see Supplementary Materials) and the reverse triangle inequality. Finally, by the triangle inequality, with probability of at least $1 - \delta$, we have:

$$\left| \rho(T) - \max(Y)^{\frac{1}{K}} \right| \leq \left| \rho(T) - \sup_{x : \|x\|=1} \left\| T^K x \right\|^{\frac{1}{K}} \right| + \left| \sup_{x : \|x\|=1} \left\| T^K x \right\|^{\frac{1}{K}} - \max(Y)^{\frac{1}{K}} \right| \leq \epsilon. \tag{16}$$

$\square$

**Remark.** *According to [32], since the optimal interface values are Turing-computable, if the depth of the GNN is at least the diameter of the graph, and a TAGConv layer followed by a feature encoder is Turing-complete, the optimal interface values can be learned. In our setting, for a problem on a structured grid of size $N \times N$ with two identical rectangular subdomains, this implies that the GNN will be able to learn the optimal interface values given, if and only the GNN has depth at least $2N$, has deep enough feature encoders, and the width of the layers is unbounded.*

**Remark.** *Theorem 1 guarantees convergence of the loss function to the spectral radius, in the limits of many samples and many stationary iterations. To the best of our knowledge, such a guarantee is not known for the previous loss functions used in the area [12, 11]. Moreover, there are substantial improvements in the numerical results using the new loss function in comparison to that of [12], as shown in Figure 6.*

**Theorem 2.** *Assuming bounded subdomain size, the time complexity to evaluate the optimal Schwarz parameters using our method is $O(n)$, where $n$ is the number of nodes in the grid.*

*Proof.* Given bounded Lloyd subdomain size and fixed number of Lloyd aggregation cycles, subdomain generation has $O(n)$ time complexity [28] (see the Supplementary Material). To evaluate each TAGConv layer, one computes $y = \sum_{\ell=1}^{L} G_\ell x_\ell + b \mathbf{1}_n$, where $L$ is the number of node features, $G_\ell \in \mathbb{R}^{n \times n}$ is the graph filter, $b$ is a learnable bias, and $x_\ell \in \mathbb{R}^n$ are the node features. Moreover, the graph filter is a polynomial in the adjacency matrix $M$ of the graph: $G_\ell = \sum_{j=0}^{J} g_{\ell,j} M^j$ where $J$ is a constant and $g_{\ell,j}$ are the coefficients of filter polynomial. Since the graph has bounded node degrees, it implies that $M$ is sparse and the action of $M^j$ has $O(n)$ cost, and therefore, the full TAGConv evaluation also has $O(n)$ cost. Moreover, the cost of edge feature and the feature networks are $O(n)$, resulting in overall $O(n)$ cost. $\square$

### 3.3 Training Details

We use a graph neural network based on four TAGConv layers and ResNet node feature encoders consisting of eight blocks after each TAGConv layer; see Supplementary Materials for more details on the structure of the GNN, and Section 4.3 for an ablation study. The training set in our study consists of 1000 unstructured grids with piecewise linear finite elements and with grids ranging from 90–850 nodes (and an average of 310 nodes). The grids are generated by choosing either a regular grid (randomly selected 60% of the time) or a randomly generated convex polygon; pygmsh [33] is used to generate the mesh on the polygon interior. A sample of the training grids is shown in Figure 2. We train the GNN for four epochs with a mini batch size of 25 using the ADAM optimizer [34] with a fixed learning rate of $10^{-4}$. For the numerical evaluation of the loss function (10) we use $K = 4$ iterations and $m = 500$ samples. The code[1] was implemented using PyTorch Geometric [35], PyAMG [36], and NetworkX [37]. All training was performed on an 8-core i9 Macbook Pro using CPU only.

Moreover, we train two special networks for use in Section 4.1. The first is a "Brute Force" network, which is a single-layer neural network without an activation function, trained only on a structured

---

[1]All code and data for this paper is at `https://github.com/compdyn/learning-oras` (MIT licensed).

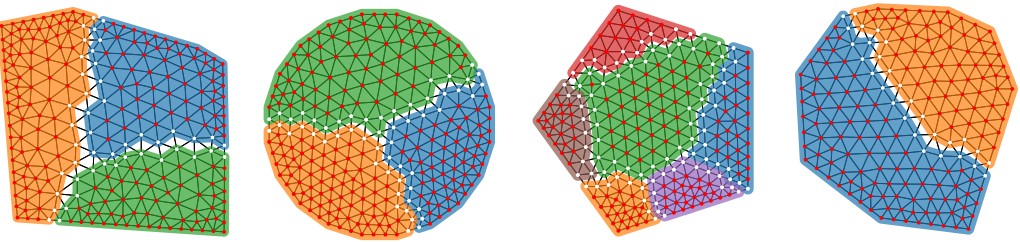

Figure 2: Example grids from the training set.

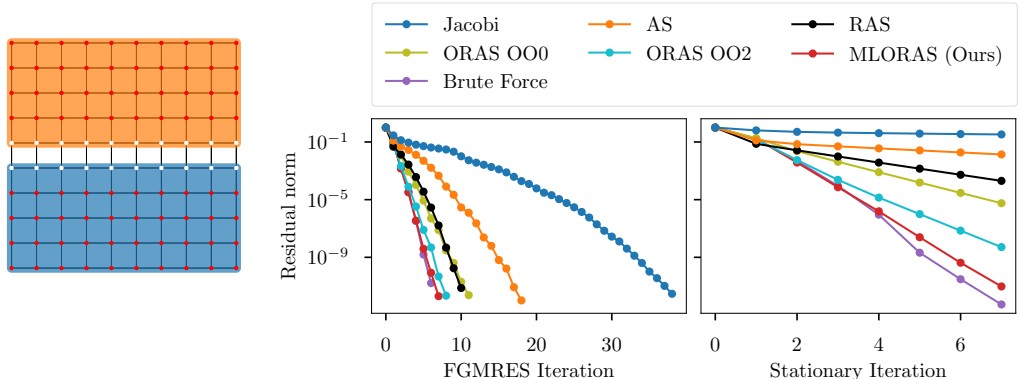

Figure 3: Results for the Helmholtz problem on a $10 \times 10$ structured grid (left) with two identical subdomains, with convergence plots for the methods used as preconditioners for FGMRES (center) and as stationary algorithms (right).

$10 \times 10$ grid with two identical subdomains using the ADAM optimizer. The purpose of training this is to obtain the optimal interface values as a benchmark for comparison, including against our method. The second is the same network used for our method, MLORAS, but overtrained on only the problem in Section 4.1, to understand the learning capabilities of the method and choice of GNN architecture.

## 4 Results

### 4.1 Two-subdomain structured grids

We first consider rectangular structured grids with two subdomains. Although restrictive, these problems allow us to directly compare to existing OSM parameters, which are only available for structured grids and exactly two subdomains. We follow [7] and consider the Helmholtz problem (1) on the unit square with $\eta = 1$ and homogeneous Dirichlet boundary conditions. We discretize the problem on an $N \times N$ rectangular grid with $h = 1/(N+1)$ grid spacing, using the standard five-point finite difference stencil.

Figure 3 shows the results of different methods on a $10 \times 10$ structured grid with two identical subdomains. We see that the overtrained MLORAS network learns interface parameters that outperform the Restrictive Additive Schwarz (RAS) [6] method; more significantly, MLORAS also outperforms the existing Optimized-RAS (ORAS) methods that were analytically derived for this specific problem (zeroth-order OO0 and second-order OO2 from [7]). Moreover, in these results, the MLORAS network almost reaches the performance of the "Brute Force" network (obtained by directly optimizing all interface values for this single structured grid), indicating that using a GNN to encode the optimal interface values does not significantly restrict the search space.

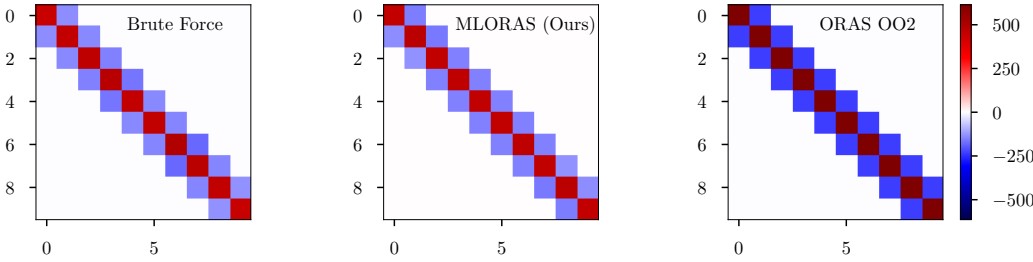

Figure 4: Interface values for the $10 \times 10$ structured grid with two identical subdomains. From left to right: brute force optimization of the interface values, MLORAS, and second-order ORAS (OO2) [7].

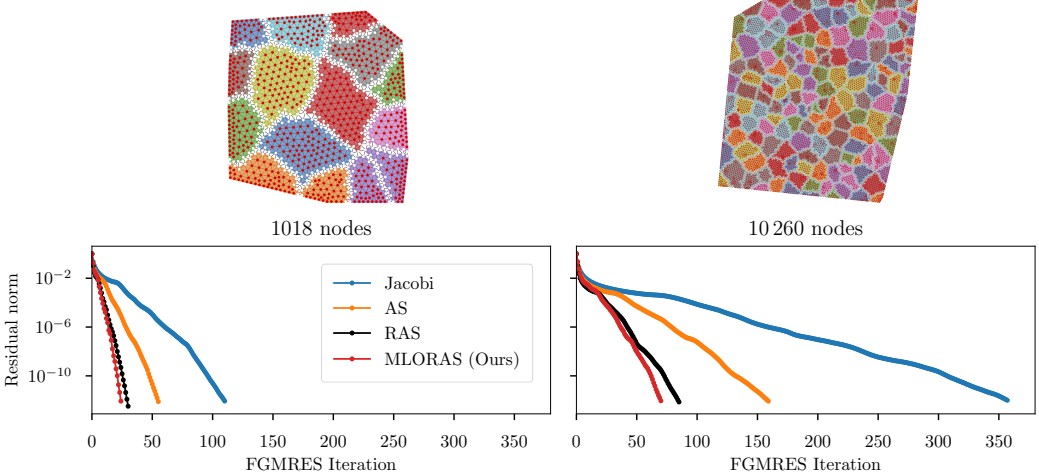

Figure 5: Example convergence on smaller (left) and larger (right) unstructured grids.

To understand the performance of the methods in more detail, Figure 4 plots the interface values output by the brute force optimization, the MLORAS network, and the OO2 ORAS algorithm. This shows that the MLORAS network is choosing interface values very close to those directly optimized by the brute force method, unlike those selected by the OO2 method.

## 4.2 Unstructured grids

To evaluate the performance of the MLORAS network on unstructured grids, we consider 16 unstructured triangular grids in 2D with sizes ranging from about 90 to 40 000 nodes. These grids are defined on convex subsets of $(0, 1) \times (0, 1)$; we solve the Helmholtz problem (1) on these domains with $\eta = 1$ and homogeneous Dirichlet boundary conditions, discretized with piecewise-linear finite elements.

Example convergence plots are shown in Figure 5, where our method (MLORAS) is compared to RAS (Restricted Additive Schwarz [6]), AS (Additive Schwarz), and Jacobi methods as a preconditioner for FGMRES. Here we see that the MLORAS network is able to learn optimized interface parameters for unstructured grids that outperform RAS, and that MLORAS can scale to problems that are much larger than those in the training set, which are all below $n = 1000$ nodes. Importantly, MLORAS retains an advantage over RAS even as the grid size increases.

Figure 6 shows the performance of three different methods across all unstructured test grids. The three methods are: (1) "F-norm optimized": the same network as MLORAS, but trained to directly optimize the Frobenius norm of the error propagation matrix, $\|A\|_F$, (2) the RAS [6] method, and (3) the MLORAS method. We do not show ORAS results here, as it cannot be applied to unstructured grids. Figure 6 reveals two important facts. First, MLORAS consistently outperforms RAS over the entire testing set, both as an FGMRES preconditioner and as a stationary algorithm, and this

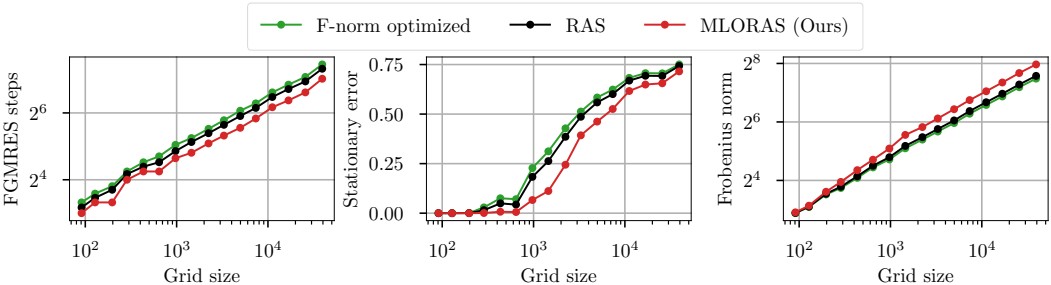

Figure 6: Convergence on all unstructured grids in the testing set. Left: the number of preconditioned FGMRES steps required to solve the problem to within a relative error of $10^{-12}$. Center: error reduction by the stationary iteration after 10 iterations. Right: Frobenius norm for each method on each test problem.

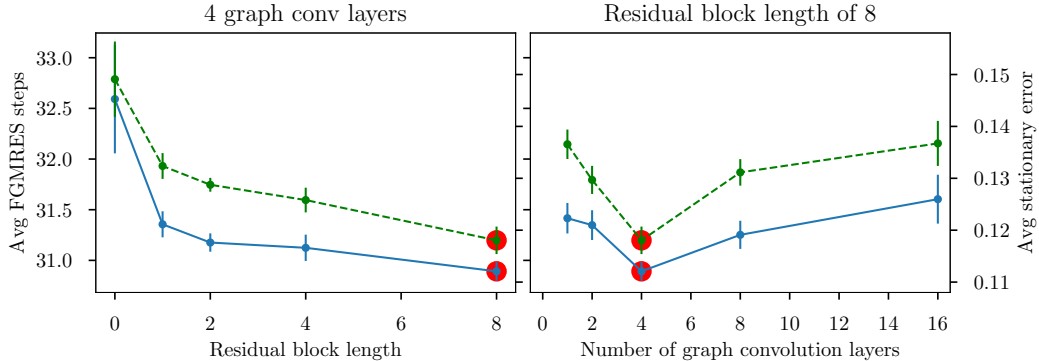

Figure 7: Ablation study results. The left panel varies the residual block length while keeping the number of TAGConv layers fixed at 4. The right panel varies the number of TAGConv layers while keeping the residual block length fixed at 8. The solid blue lines (left axis) show the average number of FGMRES steps needed to reduce the relative error below $10^{-12}$, while the dashed green lines (right axis) show the average stationary algorithm error reduction after 10 iterations. The red circles mark the results for the network with the best performance (residual block length of 8 and 4 TAGConv layers), which was used for all other studies in this paper. Error bars show one standard error of the mean.

advantage is maintained even for large grids. Second, we see that the Frobenius norm is indeed a worse choice for the loss function than our new loss in (6). We see this from the fact that MLORAS has a worse Frobenius norm than either of the other two methods, but it has the best convergence rate. In addition, when we explicitly optimize the Frobenius norm (the "F-norm optimized" method), we see that we do obtain the lowest Frobenius norm, but this translates to the worst convergence rate.

### 4.3 Ablation study

To understand the impact of the network structure on performance, we conduct an ablation study by varying the ResNet block length and the number of TAGConv layers. In each case, we train a network on five different training sets, each consisting of 1000 grids generated as described in Section 3.3. The trained networks are tested on 50 unstructured grids, each with 2400 to 2600 nodes. For each architecture, the mean performance is computed, together with error bars from the fivefold repetition, with results shown in Figure 7. We see that a higher residual block length is always better, but that the optimal number of TAGConv layers is 4, and using more layers actually decreases performance. This phenomenon has been observed in other contexts (e.g., [38] and [39]) and can be attributed to over-smoothing in GNNs. The use of residual blocks in our architecture is, thus, important to allow us to increase network depth without needing to increase the number of GNN layers.

# 5    Conclusion

We propose an unsupervised learning method based on Graph Convolutional Neural Networks (GC-NNs) for extending optimized restricted additive Schwarz (ORAS) methods to multiple subdomains and unstructured grid cases. Our method is trained with a novel loss function, stochastically minimizing the spectral radius of the error propagation operator obtained using the learned interface values. The time complexity of evaluating the loss function, as well as obtaining the interface values using our neural network are all linear in problem size. Moreover, the proposed method is able to outperform ORAS, both as a stationary algorithm and preconditioner for FGMRES on structured grids with two subdomains, as considered in the conventional ORAS literature. On more general cases, such as unstructured grids with arbitrary subdomains of bounded size, our method outperforms RAS consistently, both as a stationary algorithm and preconditioner for FGMRES. The main limitations of the current work are that the method was studied for two PDE cases, namely the Helmholtz problem in the main paper and the non-uniform Poisson problem in Appendix D. We also defer the study of nonlinear or time-dependent PDEs to future work.

## Acknowledgement

The work of S.P.M. was partially supported by an NSERC Discovery Grant. The authors thank the referees for their insightful comments. The authors have no competing interests to declare.

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
