## Appendix A  Useful lemma

This lemma is useful in the proof of Theorem 1. While its proof is quite simple, we are not aware of the result in the literature, and include it here for completeness.

**Lemma 3.** *For $x, y \in \mathbb{R}$, with $0 \leq y \leq x$ and any $K \in \mathbb{N}$, $x^{\frac{1}{K}} - y^{\frac{1}{K}} \leq (x - y)^{\frac{1}{K}}$*

*Proof.* Since $y \leq x$, the binomial theorem gives that

$$x \leq (x - y) + y + \sum_{i=1}^{K-1} \binom{K}{i} y^{\frac{i}{K}} (x - y)^{\frac{K-i}{K}} = (y^{\frac{1}{K}} + (x - y)^{\frac{1}{K}})^K. \tag{17}$$

Taking the $K^{\text{th}}$ root of both sides and rearranging gives the stated result. $\qquad\square$

## Appendix B  Solution plots

We have compared the solution plot of our method with RAS for the Holmholtz problem. We consider a $100 \times 100$ structured grid on the $(0, 1) \times (0, 1)$ domain, and consider the following true solution for the problem: $u^* = \sin(8\pi x) + \sin(8\pi y)$. We then start with an initial random guess with $L_2$ norm of 1. We apply 10 iterations of RAS and MLORAS (ours) as stationary algorithm and obtain solution for each method. Moreover, we run 10 iterations of FGMRES with MLORAS and RAS preconditioners on the initial guess and obtain predictions for both methods. The results are shown in Figure 8 for MLORAS and Figure 9 for RAS.

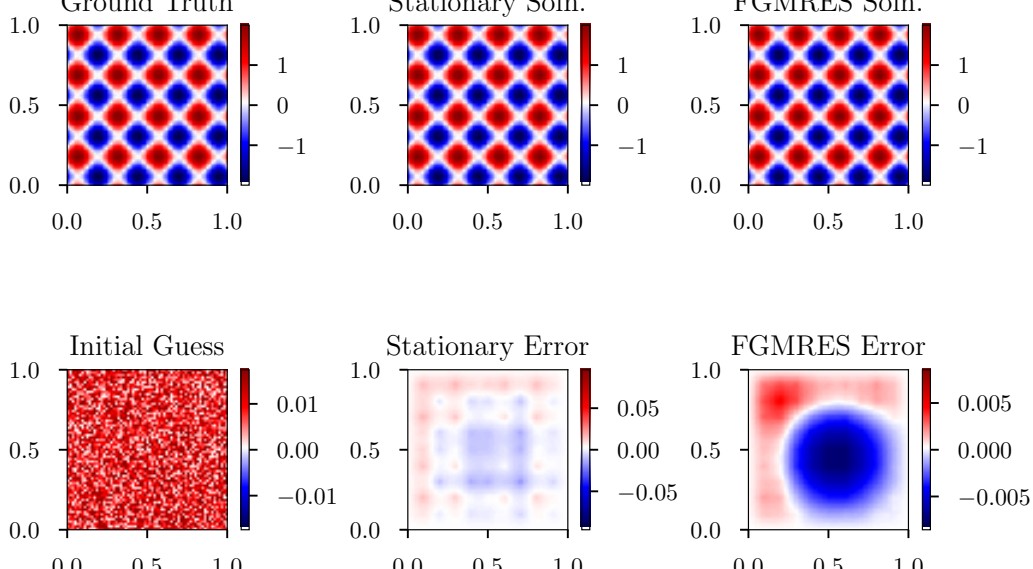

Figure 8: MLORAS (ours) solution plots. Top left: ground truth, top middle: MLORAS stationary solution after 10 iterations, top right: FGMRES solution with MLORAS preconditioner after 10 steps, bottom left: initial guess, bottom middle: error of MLORAS stationary solution ($L_2$ norm of error = 0.231), bottom right: error of FGMRES with MLORAS preconditioner solution ($L_2$ norm of error = 0.084).

## Appendix C  Neural network

### C.1  Inputs and Output

**Inputs:**  The GNN takes the grid $G$ as its input which has three main components, namely $D_{\text{node}}$ (node feature matrix), $D_{\text{edge}}$ (edge feature matrix), and $A$ (the graph adjacency matrix). Every node

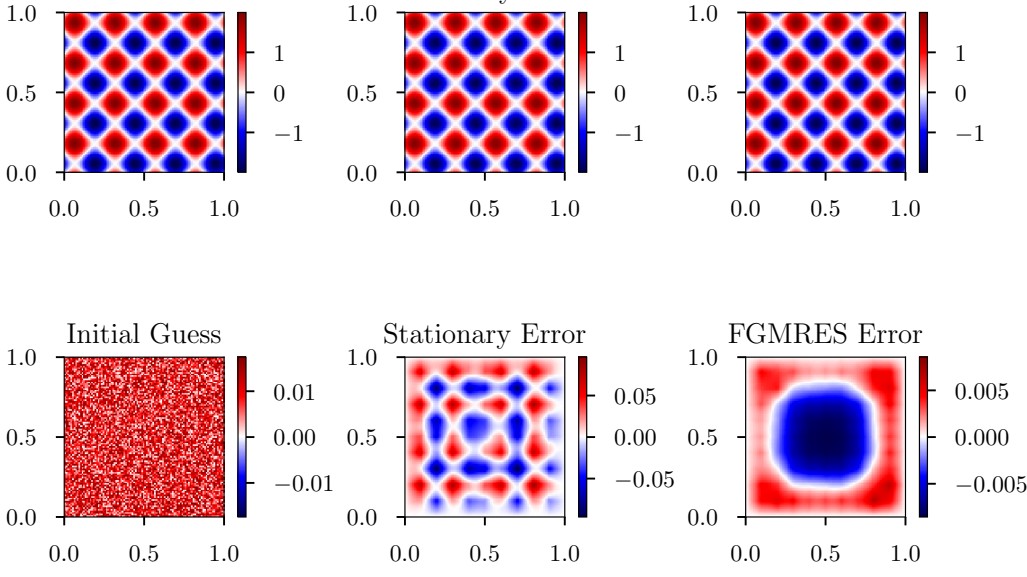

Figure 9: RAS solution plots. Top left: ground truth, top middle: RAS stationary solution after 10 iterations, top right: FGMRES solution with RAS preconditioner after 10 steps, bottom left: initial guess, bottom middle: error of RAS stationary solution ($L_2$ norm of error = 6.526), bottom right: error of FGMRES with RAS preconditioner solution ($L_2$ norm of error = 0.146).

has a binary feature, and its value is one if it is on a boundary of a subdomain and zero otherwise. Therefore, for every node, the corresponding element in $D_{node}$ indicates whether that point is on a boundary or not. In other words, the binary node feature determines the grid decomposition. $D_{edge}$ consists of the edge values obtained from discretization of the underlying PDE.

**Outputs:**  After passing the input to the GNN architecture which consists of node and edge convolution blocks and is fully described in the following subsection, the learned edge weights are obtained for every edge in the grid. However, only the edges between the nodes on the subdomain boundaries (considering self-loops) are of our interest so we mask the rest of the edges. Therefore, the output of the GNN is the learned values for edges connecting nodes on the boundary of a subdomain which are referred to as interface values in the paper. For the $i$-th subdomain, the interface value matrix is referred to as $L_i^\theta$, where $\theta$ represents the GNN learnable parameters (see Equation 7). Figure 3 shows an example of the sparsity pattern of an $L$ matrix for each of the identical subdomains of the $10 \times 10$ structured grid in Figure 4. Also note that the interface values are the nonzero elements of the corresponding $L$ matrix.

### C.2 Architecture

The overall architecture of the GNN is shown in Figure 10. The GNN takes a graph as its input and sends node and edge features to the node convolution and edge feature preprocessing blocks, respectively, both of which are shown in Figure 11. Each node convolution block consists of a TAGConv layer with 2-size filters and 128 hidden units, followed by a ReLU activation, an instance norm layer, and a feature network block. The feature network block, as shown in Figure 12, consists of 8 blocks with residual connections between each; furthermore, each of the blocks consists of a layer norm followed by a fully connected layer of size 128, followed by a ReLU activation, and finally another fully connected layer of size 128. The edge feature preprocessing block takes the edge features and applies fully connected layers, ReLU nonlinearities, and instance normalization, before the graph convolution pass.

After passing through all of the node convolution blocks, the edge and node features are sent to a stack block. This block simply stacks node features onto the edges adjacent to each node. For every edge,

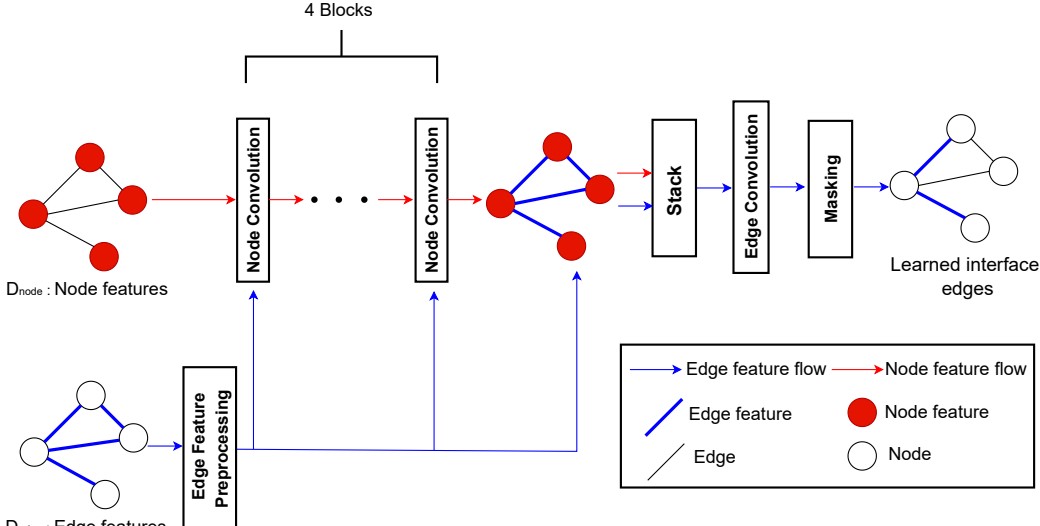

Figure 10: Overall GNN architecture.

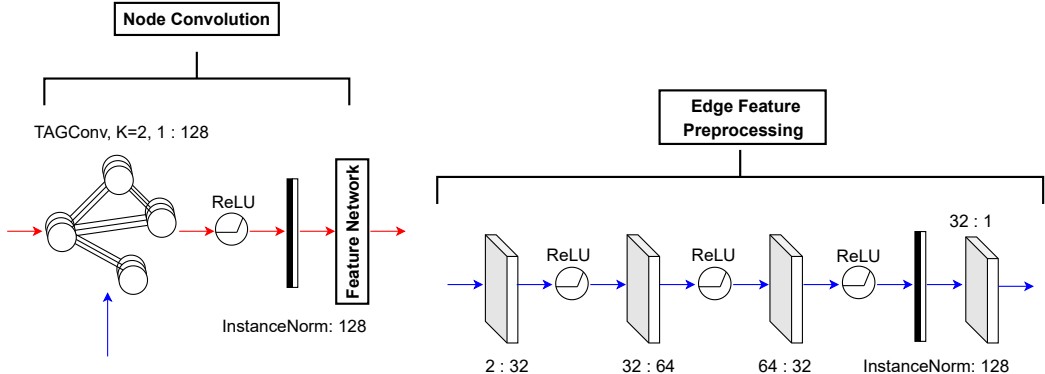

Figure 11: Left: Node convolution block. Right: Edge feature preprocessing block.

$(u, v)$, where $u$ and $v$ are the nodes on that edge, denote the node and edge features by $E_u, E_v, E_{(u,v)}$, which are the inputs to the stack block. The block then stacks these features and outputs them as the new edge features for the edge $(u, v)$. Following the stack layer is the edge convolution block, depicted in Figure 13, which takes the stacked edge and node features and passes them through a series of fully connected layers, ReLU activation functions, and layer norms. It is noteworthy that the size of the input to the edge convolution block is 257, since following the description of the stack block, the two node features, each of size 128, and the edge feature of size 1 are stacked together. The output from the edge convolution block is, finally, passed through a masking block that outputs the interface values. This masks the interior edge values, i.e., takes the output of the GNN (one value per each edge in the graph) and multiplies those values that are not on the boundary of a subdomain by zero, to restrict the output from the GNN to the desired edges in the graph.

## Appendix D   Poisson problem with discontinuous diffusion coefficient

We also consider the 2D Poisson problem with discontinuous diffusion coefficient which is formulated as follows:

$$-\boldsymbol{\nabla} \cdot \kappa(x,y)\nabla u = f \quad \text{in } \Omega, \quad \kappa(x,y) = \begin{cases} 1000 & 0 < x < 0.5 \\ 1 & 0.5 \le x < 1. \end{cases} \tag{18}$$

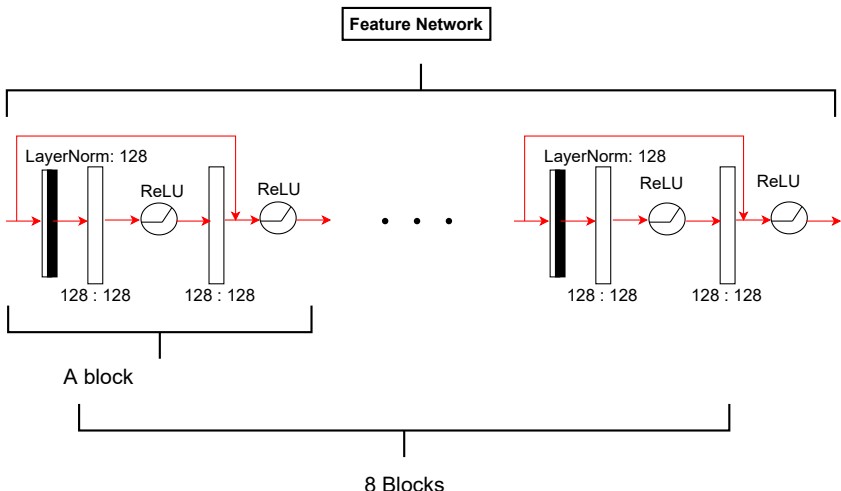

Figure 12: Feature network blocks.

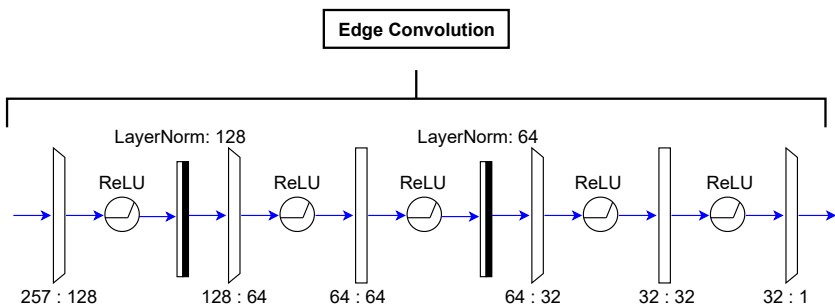

Figure 13: Edge convolution block.

where $\Omega$ is, as before, defined as a convex subset of $(0,1) \times (0,1)$ and $\kappa(x,y)$ is the discontinuous diffusion coefficient. For this problem, we consider nine domains with unstructured triangular grids, with sizes ranging from about 100 to over 30k nodes. The subdomains are generated using Lloyd aggregation (fully explained in Appendix E), with a fixed ratio of 0.015. We note that the subdomains are not constrained in any way, and a single subdomain may contain parts of the domain with different diffusion coefficients. The results of our method compared with RAS baseline for both the stationary algorithm and the FGMRES preconditioner are shown in Figure 14, and show little qualitative difference with the earlier results for Helmholtz.

## Appendix E  Lloyd aggregation

Lloyd's algorithm [29] is a standard approach for partitioning data, closely related to $k$-means clustering, that can be used (for example) to find close approximations to centroidal Voroni tesselations. Here, we use a modified form of Lloyd's algorithm, known as Lloyd aggregation [28], to partition a given set of degrees of freedom, $V$, into the non-overlapping subdomains, $V_i^0$, for $i \in \{1, 2, \ldots, S\}$, needed as input to our algorithm. (Note that, here, we change notation from the paper and use $V$ to denote the vertices in the graph associated with the matrix, rather than $D$ to denote the index set of degrees of freedom.) Consider a 2D planar graph, $G$, the set of all edges $E$, the set of all of its nodes $V$, and $V_c \subseteq V$. The nodes in $V_c$ serve as the centers of "clusters" in the graph that will define the subdomains, $V_i^0$. These regions are obtained based on the closest center to each graph node, where the distance is measured by the number of edges covered in the shortest path between two nodes (denote distance in the graph between node $i$ and $j$ by $g_{ij}$). Define the centroid of a region as the farthest node from the boundary, breaking possible ties by random choice. We use a modified version of the Bellman-Ford algorithm, commonly used for obtaining the nearest center to every node in

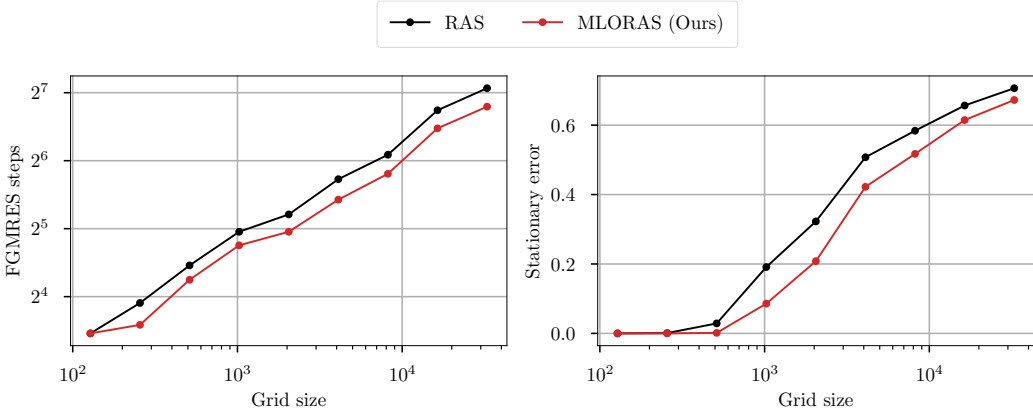

Figure 14: Discontinuous diffusion coefficient for Poisson problem on various size grids. Left: the number of preconditioned FGMRES steps required to solve the problem to within a relative error of $10^{-12}$. Right: error reduction by the stationary iteration after 10 iteration.

$V$ and its associated distance [40]. Let $\vec{n}$ be a list of graph nodes whose $i$-th element is the nearest center to the $i$-th node of the graph, and let $g_j$ be the graph distance from node $j$ to $n_j$; then, the modified Bellman-Ford algorithm is shown in Algorithm 1.

---

**Algorithm 1** Modified Bellman-Ford

---

1: **Input** $E$: The set all edges, $V$: The set of all nodes, $V_c$: The set of initial center nodes.
2: $g_i = \infty \ \forall_{i \in \{1,2,...,|V|\}}$
3: $n_i = -1 \ \forall_{i \in \{1,2,...,|V|\}}$
4: **for** $c \in V_c$ **do**
5: $\quad g_c \leftarrow 0$
6: $\quad n_c \leftarrow c$
7: **end for**
8: **while** True **do**
9: $\quad$ Finished $\leftarrow$ True
10: $\quad$ **for** $(i,j) \in E$ **do**
11: $\quad\quad$ **if** $g_i + g_{ij} < g_j$ **then**
12: $\quad\quad\quad g_j \leftarrow g_i + g_{ij}$
13: $\quad\quad\quad n_j \leftarrow n_i$
14: $\quad\quad\quad$ Finished $\leftarrow$ False
15: $\quad\quad$ **end if**
16: $\quad$ **end for**
17: $\quad$ **if** Finished **then**
18: $\quad\quad$ **return** $\vec{g}, \vec{n}$
19: $\quad$ **end if**
20: **end while**

---

While this is an iterative computation, it has finite termination when the values in $\vec{g}$ and $\vec{n}$ stop changing. After running this modified Bellman-Ford algorithm, Lloyd's algorithm modifies the clusters by selecting the centroid of every subdomain as its new center, then iterates, using the modified Bellman-Ford algorithm to calculate new distances and nearest centers. Given updated center positions, it forms the new subdomains. The full Lloyd algorithm is shown in Algorithm 2, where we define the set of border nodes, $B$, as the set of all nodes that are connected by an edge to a node that has a different nearest center node. The key point here is that we use Modified Bellman-Ford to assign closest centers, then compute the set of border nodes, then find the new centers as those that are further from the border set within each of the original subdomains (using graph distances from $B$, but original center assignment in $\vec{n}$).

**Algorithm 2** Lloyd Aggregation

1: **Input** $K$: Number of iterations, $E$: The set of all edges, $V$: The set of all nodes, $V_c$: The set of initial center nodes.
2: **for** $i = 1, 2, 3, ..., K$ **do**
3:     $\vec{g}, \vec{n} \leftarrow$ Modified Bellman-Ford$(E, V, V_c)$
4:     $B \leftarrow \emptyset$
5:     **for** $(i, j) \in E$ **do**
6:         **if** $n_i \neq n_j$ **then**
7:             $B \leftarrow B \cup \{i, j\}$
8:         **end if**
9:     **end for**
10:     $\vec{g}, \vec{x} \leftarrow$ Modified Bellman-Ford$(E, V, B)$
11:     $V_c \leftarrow \{i \in V : g_i > g_j \;\; \forall n_i = n_j\}$
12: **end for**
13: **return** $\vec{n}$

**Time Complexity:**  Assuming each node's initial distance to a center node is bounded independently of $|V|$, and also assuming that each node's degree is bounded independently of $|V|$, Algorithm 1 runs a $|V|$-independent number of iterations to determine one nearest center node for every point. Thus, Algorithm 1 is $O(|V|)$ in our case. This is run a $|V|$-independent number of times in Algorithm 2, to generates the subdomains, resulting in an overall algorithmic cost of $O(|V|)$.