# OpenReview forum: "Learning Interface Conditions in Domain Decomposition Solvers"
_NeurIPS.cc/2022/Conference — NeurIPS 2022 Accept_

### Official Review · Reviewer_4BC2 · 2022-07-07

**Rating:** 6
**Confidence:** 3
**Soundness:** 3 good
**Presentation:** 3 good
**Contribution:** 2 fair

**Summary:**

This paper presents a learning-based approach to find the optimal boundary values for domain-decomposed PDE solvers. The major contribution is the application of TAGConv to solving PDEs, and a new numerical method to compute the loss defined on the spectral radius of the difference matrix.

**Questions:**

None. The paper is well-written and comprehended.

**Limitations:**

The author only tested their method on Helmholtz equation, which is the major limitation. But the authors also have mentioned this limitation and further validation remains for future works.

**Strengths And Weaknesses:**

The method looks solid, and is well-validated. The results look promising, and the application of TAGConv to domain-decomposition is novel.

However, the paper doesn't have much contribution to machine learning itself, and thus I'm not sure it fits the scope of NeurIPS. I would recommend the author resubmit this work to a journal for numerical PDE solvers.

---

> ### Author Response · Authors · 2022-08-02
> **Our response**
>
> We thank the reviewer for their clear and concise review. The reviewer raises a valid point about where to present work at the intersection of two fields (in this case machine learning and numerical analysis). In this case, machine learning conferences have become the standard venue for presenting intersectional work on PDE solvers and neural networks (e.g., [1] at NeurIPS 2021, [2,3] at NeurIPS 2020, [4] at ICML 2020, [5] at ICML 2019, [6] at ICLR 2021, etc.). We are continuing in this tradition to make our work accessible to the research community in this area. We also believe that NeurIPS is the correct venue for our current paper because numerical analysis venues do not generally have reviewers with adequate background knowledge to assess the work. Finally, we note that the NeurIPS Call for Papers explicitly invites submissions on “Machine Learning for Sciences”, which we think includes the current work.
>
> While simple, the Helmholtz equation is the definitive prototypical system that has been used by most of previous work in this area [7-9], which is why we also focused our attention on it.  In particular, there are existing theoretical results concerning the convergence behavior of optimized Schwarz methods on structured grids with two subdomains, and solving the Helmholtz equation allows us to compare our method to the previous work in this area and demonstrate that learning-based methods give superior solvers [Figure 3].  Nonetheless, as discussed below, we propose to add some additional results to the manuscript, for some generalizations of this problem, where our technique continues to show improved performance.
>
> While the focus on the Helmholtz equation is, as explained above, motivated by the desire to compare to existing results in the optimized Schwarz literature, we recognize the need to test a wider range of problems.  As such, we propose to add the following numerical examples to the manuscript (as supplemental material).
>
> We also consider the 2D Poisson problem with discontinuous diffusion coefficient which is formulated as Equation 18, where $\Omega$ is, as before, defined as a convex subset of $(0,1)\times(0,1)$ and $\kappa(x,y)$ is the discontinuous diffusion coefficient. For this problem, we consider nine domains with unstructured triangular grids, with sizes ranging from about 100 to over 30k nodes. The subdomains are generated using Lloyd aggregation (fully explained in Appendix D), with a fixed ratio of 0.015. We note that the subdomains are not constrained in any way, and a single subdomain may contain parts of the domain with different diffusion coefficients. The results of our method compared with RAS baseline for both the stationary algorithm and the FGMRES preconditioner are shown in Figure 12, and show little qualitative difference with the earlier results for Helmholtz.
>
>
>
>
>
>
>
>
> [1] Taghibakhshi, A., MacLachlan, S., Olson, L. and West, M., 2021. Optimization-based algebraic multigrid coarsening using reinforcement learning. Advances in Neural Information Processing Systems, 34, pp.12129-12140.
>
> [2] Li, Z., Kovachki, N., Azizzadenesheli, K., Liu, B., Stuart, A., Bhattacharya, K. and Anandkumar, A., 2020. Multipole graph neural operator for parametric partial differential equations. Advances in Neural Information Processing Systems, 33, pp.6755-6766.
>
> [3] Um, K., Brand, R., Fei, Y.R., Holl, P. and Thuerey, N., 2020. Solver-in-the-loop: Learning from differentiable physics to interact with iterative pde-solvers. Advances in Neural Information Processing Systems, 33, pp.6111-6122.
>
> [4] Luz, I., Galun, M., Maron, H., Basri, R. and Yavneh, I., 2020, November. Learning algebraic multigrid using graph neural networks. In International Conference on Machine Learning (pp. 6489-6499). PMLR.
>
> [5] Greenfeld, D., Galun, M., Basri, R., Yavneh, I. and Kimmel, R., 2019, May. Learning to optimize multigrid PDE solvers. In International Conference on Machine Learning (pp. 2415-2423). PMLR.
>
> [6] Zongyi Li, Nikola Kovachki, Kamyar Azizzadenesheli, Burigede Liu, Kaushik Bhattacharya, Andrew Stuart, and Anima Anandkumar. Fourier neural operator for parametric partial differential equations. International Conference on Learning Representations (ICLR 2021).
>
> [7] Gander, M.J. and Kwok, F., 2011. Optimal interface conditions for an arbitrary decomposition into subdomains. In Domain Decomposition Methods in Science and Engineering XIX (pp. 101-108). Springer, Berlin, Heidelberg.
>
> [8] Amik St-Cyr, Martin J Gander, and Stephen J Thomas. Optimized multiplicative, additive, and restricted additive Schwarz preconditioning. SIAM Journal on Scientific Computing, 29(6):2402– 2425, 2007.
>
> [9] M.J. Gander, L. Halpern, and F. Nataf. Optimized Schwarz methods. In Proceedings of the 12th International Conference on Domain Decomposition, pages 15–27. ddm.org, 2000

---

### Official Review · Reviewer_H1jk · 2022-07-10

**Rating:** 6
**Confidence:** 3
**Soundness:** 3 good
**Presentation:** 3 good
**Contribution:** 2 fair

**Summary:**

Domain decomposition solvers are effective for large-scale PDE problems and optimized Schwarz domain decomposition methods are commonly used for structured-grid problems. In this work, the authors proposed to learn the parameters in Schwarz preconditioner that generalize it to unstructured grid and enable faster convergence. The main difference compared to the prior work, which works with a different preconditioner, is the proposed new loss function inspired by Gelfand's formula; and a practical approximation of the loss function via stochastic sampling.  Experiments on the 2D Helmholtz equation have been done to demonstrate the effectiveness of the approach.

**Questions:**

1. how are boundary nodes handled for the input graph to the GNN?
2. it's a bit unclear to be for the input/output of the GNN and how is that done. In line 139, it is noted that the input $i$ the $D$ (total domain) and its decomposition, and the sparsity constraint $L_i$, and predict the matrices. How exactly the "decomposition", and "sparsity constraints" are encoded in the input graph? According to the supplement, the output is the "learned interface edge features" and that will give us the predicted matrices, is this understanding correct?

**Ethics Review Area:**

["I don’t know"]

**Limitations:**

There is only one experiment setting with the Helmholtz equation. It would be more appealing if we have another experiment with a different PDE and boundary condition.


**Strengths And Weaknesses:**

originality - the proposed loss function is different from the prior works and the stochastic sampling technique used to approximate the loss function is new to be applied to this problem, to the best of my knowledge.

quality/clarity - the figures are clear and the article is well-written. More detailed captions could be added to facilitate reading. It would be nice to add a list of contributions by the end of the introduction to make the contributions clear.

significance - the proposed approach demonstrates its effectiveness in the 2D Helmholtz equation problem. The approach scale linearly with the problem made is potentially be used for larger-scale problems.

---

> ### Author Response · Authors · 2022-08-02
> **Our response**
>
> Our Response:
> We thank the reviewer for their helpful clarifying questions, which will help to improve our paper.
> In response to questions:
>
> 1- When we input the graph to the GNN, every node has a binary feature, and its value is one if it is a boundary node and zero otherwise. This node feature matrix has the information needed to distinguish boundary nodes as an input to the GNN. We will add text regarding this to the first paragraph of Appendix B in the supplementary materials to further clarify this.
>
> 2- The inputs to the GNN are the grid adjacency matrix, the edge weights (which come from the discretization of the PDE), the grid decomposition into subdomains, and a boundary marker (which is encoded as a binary feature for every node, denoting whether or not it is on the boundary of some subdomain).  As shown in Figure 8 in the supplementary materials, after passing through node and edge convolution blocks, edge weights are obtained for every edge, but we mask the edges that are not between the boundary nodes.  Furthermore, in postprocessing, we take the output of the GNN (the learned values for all edges connecting boundary nodes in any subdomain (referred to as interface values in the paper)) and separate this into its contribution to each individual matrix, $L_i$.  We will add these clarifications to Appendix B in the supplementary materials, where we describe the GNN architecture.  Similar changes will be made to the discussion in Section 2, and around Figure 4, to clarify the construction of $L_i$, as requested by reviewer 7Eu3.
>
> Limitations:
>
> While simple, the Helmholtz equation is the definitive prototypical system that has been used by most of previous work in this area [1-5], which is why we also focused our attention on it.  In particular, there are existing theoretical results concerning the convergence behavior of optimized Schwarz methods on structured grids with two subdomains, and solving the Helmholtz equation allows us to compare our method to the previous work in this area and demonstrate that learning-based methods give superior solvers [Figure 3].  Nonetheless, as discussed below, we propose to add some additional results to the manuscript, for some generalizations of this problem, where our technique continues to show improved performance.
>
> While the focus on the Helmholtz equation is, as explained above, motivated by the desire to compare to existing results in the optimized Schwarz literature, we recognize the need to test a wider range of problems.  As such, we propose to add the following numerical example to the manuscript (as supplemental material).
>
> We consider the 2D Poisson problem with discontinuous diffusion coefficient which is formulated as Equation 18, where $\Omega$ is, as before, defined as a convex subset of $(0,1)\times(0,1)$ and $\kappa(x,y)$ is the discontinuous diffusion coefficient. For this problem, we consider nine domains with unstructured triangular grids, with sizes ranging from about 100 to over 30k nodes. The subdomains are generated using Lloyd aggregation (fully explained in Appendix D), with a fixed ratio of 0.015. We note that the subdomains are not constrained in any way, and a single subdomain may contain parts of the domain with different diffusion coefficients. The results of our method compared with RAS baseline for both the stationary algorithm and the FGMRES preconditioner are shown in Figure 12, and show little qualitative difference with the earlier results for Helmholtz.
>
> References:
>
> [1] Gander, M.J. and Kwok, F., 2011. Optimal interface conditions for an arbitrary decomposition into subdomains. In Domain Decomposition Methods in Science and Engineering XIX (pp. 101-108). Springer, Berlin, Heidelberg.
>
> [2] Amik St-Cyr, Martin J Gander, and Stephen J Thomas. Optimized multiplicative, additive, and restricted additive Schwarz preconditioning. SIAM Journal on Scientific Computing, 29(6):2402– 2425, 2007.
>
> [3] M.J. Gander, L. Halpern, and F. Nataf. Optimized Schwarz methods. In Proceedings of the 12th International Conference on Domain Decomposition, pages 15–27. ddm.org, 2000
>
> [4] Gong, S., Gander, M.J., Graham, I.G. and Spence, E.A., 2021. A variational interpretation of Restricted Additive Schwarz with impedance transmission condition for the Helmholtz problem. arXiv preprint arXiv:2103.11379.
>
> [5] Gong, S., Gander, M.J., Graham, I.G., Lafontaine, D. and Spence, E.A., 2021. Convergence of parallel overlapping domain decomposition methods for the Helmholtz equation. arXiv preprint arXiv:2106.05218.

---

> > ### Comment · Reviewer_H1jk · 2022-08-09
> > **Thank you**
> >
> > Thanks for the clarification and additional experiments.

---

### Official Review · Reviewer_7Eu3 · 2022-07-12

**Rating:** 6
**Confidence:** 4
**Soundness:** 3 good
**Presentation:** 2 fair
**Contribution:** 3 good

**Summary:**

Learning Interface Conditions in Domain Decomposition Solvers

In this work, the authors use machine learning tools to enhance the domain decomposition solvers. It uses a graph convolutional network to learn the preconditioner. Specifically, the GNN takes the domain D and its decomposition as inputs, and outputs the boundary matrices L.

Numerically, the paper mainly considers a Helmholtz equation with constant coefficient and inhomogeneous boundary conditions. The experiments show the proposed method has a better convergence rate compared to the state-of-the-art solvers.

**Questions:**

There could be more definitions or an example of the matrices L. What is its shape and what do the entries represent?


**Limitations:**

The authors discuss that the study is limited to the Helmholtz equation.
Besides, in the reviewer's opinion, the decomposition method as an iterative method is limited to elliptic PDEs. It will be interesting to see how the decomposition work for time-dependent equation and discontinuous problem. Do we need to change the decomposition over time?

**Strengths And Weaknesses:**

Strengths:
- The purposed method can be used to solve problems on larger domains (>10000 nodes).
- The method is theoretical guarantees inherited from the Domain Decomposition solver.
- And it has a better convergence rate than the numerical solvers.
- Linear complexity

Weakness:

- In the reviewer’s opinion, the testing problem is somehow too simple. The paper mentioned non-linear equations, but only a linear Helmholtz equation is considered. It will be also interesting to study the time-dependent equations. Can the iterative solver be used to solve hyperbolic PDEs?
- The new loss function appears somewhat increment to the reviewer.

---

> ### Author Response · Authors · 2022-08-02
> **Our response**
>
> We thank the referee for their insightful comments.
> Response to the weaknesses:
>
> While simple, the Helmholtz equation is the definitive prototypical system that has been used by most of previous work in this area (references 4,5, and 7 from the manuscript), which is why we also focused our attention on it.  In particular, there are existing theoretical results concerning the convergence behavior of optimized Schwarz methods on structured grids with two subdomains, and solving the Helmholtz equation allows us to compare our method to the previous work in this area and demonstrate that learning-based methods give superior solvers [Figure 3].  Nonetheless, as discussed below, we propose to add some additional results to the manuscript, for some generalizations of this problem, where our technique continues to show improved performance.
>
> In general, domain decomposition methods are most effective for elliptic boundary-value problems.  They can be applied to nonlinear problems, either using their nonlinear variants, or successively solving linearizations (e.g., via Newton’s or Picard’s method), and are known to be effective in this case. Time-dependent problems are normally solved by using a time stepping algorithm in the time domain (e.g., Runge-Kutta methods); for implicit methods, these require the solution of a spatial problem for each timestep.  When these spatial systems are similar to those generated by elliptic PDEs (as, for example, in the case of parabolic time-dependent PDEs), solvers such as those developed here can also be leveraged. The current study was motivated by limitations in existing results for the optimized Schwarz class of domain decomposition methods, so we have focused our attention on the elliptic PDE case most commonly considered there.  While we propose below to include results for additional problems in this class, we defer the study of nonlinear or time-dependent PDEs to future work.  We will amend the last sentence of the conclusion, regarding future study, to make this limitation clearer, as well as adding a note in the introduction with more details.
>
> We certainly agree with the reviewer that there is (as cited in the manuscript) prior work on the question of what is an effective loss function in this setting.  While the change may seem incremental to references [12,31] from the manuscript, we do believe that it is an important step forward, for two reasons.  First of all, we are able to provide, in Theorem 1, a convergence guarantee of our loss function to the spectral radius, in the limits of many samples and stationary iterations.  This is not (to our knowledge) known for the loss functions used in [12,31].  Secondly, we see substantial improvements in our numerical results using the new loss function in comparison to that of [12], as documented in Figure 6.  Given that the new loss function shows both theoretical and practical improvements over those existing in the literature, we believe there is some inherent value in its use, which we document here.  We propose to add another remark to Section 3.2 to emphasize these advantages.
>
> Response to Questions:
> The matrix $L_i$ has the same dimensions as $A_i$, so that $\tilde{A}_i^\delta$ is well-defined.  However, it has a significantly different sparsity pattern, with nonzero entries only in rows/columns corresponding to nodes on the boundary of subdomain $D_i^\delta$.  In practice, we identify a cycle or path in the graph corresponding to $A_i$ with the property that every node in the cycle is on the boundary of $D_i^\delta$ but not the boundary of $D$ (the discretized domain), and then restrict the nonzeros in $L_i$ to the entries corresponding to the edges in this cycle/path (including self-edges, corresponding to entries on the diagonal of $L_i$).  We propose to add the above text as clarification to the paragraph following Equation (2) in Section 2 of the manuscript, as well as a clearer explanation around Figure 4 (which, we now realize, confuses this issue by only showing a submatrix of $L_i$ for this example).  If requested, we can additional examples as supplementary material, including for the case of a subdomain that intersects the boundary of $D$ (where $L_i$ is determined by a path (or set of paths) between nodes on the boundary of $D$, as in Figure 4) and the case of a subdomain that is interior to $D$, so that $L_i$ is determined by a cycle.
> In response to the limitations:
>
> While the focus on the Helmholtz equation is, as explained above, motivated by the desire to compare to existing results in the optimized Schwarz literature, we recognize the need to test a wider range of problems.  We consider the 2D Poisson problem with discontinuous diffusion coefficient which is formulated as Equation 18. The results of our method compared with RAS baseline for both the stationary algorithm and the FGMRES preconditioner are shown in Figure 12, and show little qualitative difference with the earlier results for Helmholtz.

---

> > ### Comment · Area_Chair_aawR · 2022-08-07
> > **Any feedback?**
> >
> > Dear Reviewer 7Eu3, authors have provided feedback. Any chance for "live discussion"? I find it really interesting in most of the cases.

---

> > ### Comment · Reviewer_7Eu3 · 2022-08-09
> > **response**
> >
> > Thank you for the detailed response and sorry for the delayed reply.
> >
> > Testing problem: I agree that domain decomposition methods are most effective for elliptic boundary-value problems, but to my knowledge, the elliptic equation like Helmholtz and Poisson are usually easy to solve with simple finite-difference methods (compared to hyperbolic equations such as the Navier-stokes). I vote in favor of this paper, but I just feel the result is not impressive enough for a strong acceptance. It will be interesting if the author can experiment on some harder cases, for example
> > (1) really large domain: in the paper, the authors experiment on a large problem with up to 40000 nodes, but in lithograph, people have been working on 10^7 pixels (https://arxiv.org/pdf/2207.04056.pdf)
> > (2) high frequency: Helmholtz gets hard when it has very high frequencies and small-scale waves.
> > Overall, there are many figures of the grids and graphs, but no figures showing the equation/problems (ground truth, prediction, error), which should be included, at least in the appendix.
> >
> > Notations: thanks for the clarification. Is it possible to have a formal definition of matrix L? example is also good.
> >
> > Overall, I vote in favor of this work. I will raise my score from borderline accept to weak accept.

---

> > > ### Author Response · Authors · 2022-08-09
> > > **Our response**
> > >
> > > We thank the reviewer for the additional feedback.  We will make a figure to include to show the truth/prediction/error for our method as an additional element in the supplementary materials.  We will also include the formal definition of matrix L and some examples in a future revision.  We will also explore whether including larger-scale problems is feasible - we have not yet considered this, but will include such results as we can generate in the time allowed for revision.

---

### Meta-Review · Area_Chair_aawR · 2022-08-26

**Recommendation:** Accept
**Confidence:** Less certain

**Metareview:**

The paper proposes a scheme to learn preconditioners for domain decomposition solvers. Graph neural network is used to learn interface matrices, and the training dataset consist of many unstructured grids for which optimal parameters are learning.
 The loss function is adapted from Daulbaev et.and is based on the Gelfand formula (with average replaced by maximum). Theoretical study is provided. The paper is well-presented, contains a useful and practical algorithm which potentially will be used by many researchers doing numerical simulations.

**Award:**

No

---

### Decision · Program_Chairs · 2022-09-14

Accept